# A Third Dose of the COVID-19 Vaccine, CVnCoV, Increased the Neutralizing Activity against the SARS-CoV-2 Wild-Type and Delta Variant

**DOI:** 10.3390/vaccines10040508

**Published:** 2022-03-25

**Authors:** Olaf-Oliver Wolz, Sarah-Katharina Kays, Helga Junker, Sven D. Koch, Philipp Mann, Gianluca Quintini, Philipp von Eisenhart-Rothe, Lidia Oostvogels

**Affiliations:** 1CureVac AG, Friedrich-Miescher-Straße 15, 72076 Tübingen, Germany; sven.koch@curevac.com (S.D.K.); gianluca.quintini@curevac.com (G.Q.); philipp.voneisenhart-rothe@curevac.com (P.v.E.-R.); 2CureVac AG, Schumannstr. 27, 60325 Frankfurt, Germany; sarah-katharina.kays@curevac.com (S.-K.K.); helga.junker@curevac.com (H.J.); philipp.mann@curevac.com (P.M.); lidia.oostvogels@curevac.com (L.O.)

**Keywords:** SARS-CoV-2, COVID-19 vaccine, neutralizing antibodies, delta variant

## Abstract

A third dose of CVnCoV, a former candidate mRNA vaccine against SARS-CoV-2, was previously shown to boost neutralizing antibody responses against SARS-CoV-2 wild-type in adults aged 18–60 and >60 years in a phase 2a clinical study. In the present study, we report the neutralizing antibody responses to a wild-type and a variant of concern, Delta, after a third dose of the vaccine on day (D)57 and D180. Neutralization activity was assessed using a microneutralization assay. Comparable levels of neutralizing antibodies against the wild-type and Delta were induced. These were higher than those observed after the first two doses, irrespective of age or pre-SARS-CoV-2-exposure status, indicating that the first two doses induced immune memory. Four weeks after the third dose on D180, the neutralizing titers for wild-type and Delta were two-fold higher in younger participants than in older participants; seroconversion rates were 100% for wild-type and Delta in the younger group and for Delta in the older group. A third CVnCoV dose induced similar levels of neutralizing responses against wild-type virus and the Delta variant in both naïve and pre-exposed participants, aligning with current knowledge from licensed COVID-19 vaccines that a third dose is beneficial against SARS-CoV-2 variants.

## 1. Introduction

The severe acute respiratory syndrome coronavirus 2 (SARS-CoV-2) that causes coronavirus disease 2019 (COVID-19) is responsible for the current pandemic declared by the World Health Organization (WHO) on 11 March 2020 [1]. As of 1 February 2022, there have been nearly 377 million confirmed cases of COVID-19, including more than 5.7 million deaths, reported to the WHO and nearly 9.9 billion vaccine doses have been administered worldwide [2]. SARS-CoV-2, an RNA virus, has genetically evolved over time, resulting in the emergence of variants from different geographic regions [1]. Some variants with mutations that modify the SARS-CoV-2 spike protein, which is the antigen in authorized COVID-19 vaccines to date, are more resistant to the host neutralization response [3,4]. The lineage includes three main subtypes (B1.617.1, B.1.617.2 and B.1.617.3), which contain diverse mutations in the receptor-binding domain of the SARS-CoV-2 spike protein. These mutations may increase the immune evasion potential of these variants, although results from some studies suggest that cross-neutralization can occur [5,6,7].

In May 2021, the WHO classified the Delta mutant B.1.617.2 as a variant of concern, and in early January 2022 [8], Delta was the most prevalent variant globally [5]. Delta infections can cause typical COVID-19 symptoms, but with more severe disease than that caused by wild-type SARS-CoV-2, which is mostly observed in unvaccinated individuals, although some fully vaccinated individuals have also been infected [7]. 

Recently, we reported the results from three clinical trials assessing the safety, efficacy and immunogenicity of a first-generation COVID-19 vaccine candidate: CVnCoV [9,10,11]. CVnCoV was developed as a sequence-optimized unmodified mRNA vaccine using the proprietary RNActive^®^ technology platform. The mRNA, which encodes a stabilized form of spike protein from the wild-type strain, is encapsulated in lipid nanoparticles. Clinical development of the first-generation CVnCoV vaccine has been stopped, due to the concomitant development of improved second-generation vaccine candidates [12]. 

We assessed the reactogenicity and immunogenicity in younger (18–60 years of age) and older (>60 years of age) adults after two or three 12 µg CVnCoV doses in a Phase 2a trial. The third dose was administered as per protocol either four weeks (>60 years of age) or five months (18–60 and >60 years of age) after the second dose in a subset of subjects [11]. In the present study, we report the neutralizing antibody response against the SARS-CoV-2 wild-type and Delta sub-type B.1.617.2 variant after the administration of a third (e.g., booster) dose of the CVnCoV vaccine in a subset of the phase 2a trial participants [11].

## 2. Materials and Methods

### 2.1. Study Design and Participants

Participant samples analyzed in this report are from a phase 2a randomized clinical trial (ClinicalTrials.gov NCT04515147) assessing the safety and immunogenicity of the first-generation CVnCoV vaccine [11]. In this trial, open-label subsets of participants who had received 12 µg of CVnCoV on Day (D)1 and D29, received a third dose (12 µg) on D57 (*n* = 30, aged >60 years old) or on D180 (*n* = 30, aged 18–60 years and *n* = 15, aged >60 years). Participants’ SARS-CoV-2 serostatus (naïve or pre-exposed) was determined retrospectively after enrollment in the trial, and only data from either SARS-CoV-2-naïve participants (on all time points) or pre-exposed participants at baseline are shown. 

### 2.2. Assessment of Neutralization Immune Response

Sera were collected before the administration of the third dose on D57 and D180, and four weeks later on D85 and D208, respectively. A microneutralization assay was performed in the laboratories of VisMederi Srl. (Siena, Italy) to determine 50% neutralization titers for the wild-type virus and the Delta variant [9]. The wild-type virus was obtained from the European Virus Archive Consortium (Human 2019-nCoV strain 2019-nCoV/Italy-INMI1, clade V; wild-type Wuhan strain-like) and the Delta variant (B.1.617.2) was obtained from Institute Pasteur by VisMederi Srl. 

Geometric mean titers (GMTs) of neutralizing antibodies and their 95% confidence intervals (CIs) were determined for the wild-type virus and Delta SARS-CoV-2 variant and summarized for each time point, according to age group and N-antigen serostatus (see below). Group seroconversion rates for neutralizing antibodies, defined as at least a four-fold increase in titers over the baseline and geometric mean-fold rise (GMFR) in titers, were calculated from baseline four weeks after the third dose (D85 for the D57 dose and D208 for the D180 dose).

### 2.3. Prior Exposure to the SARS-CoV-2 Virus

To determine if participants were SARS-CoV-2 naïve or pre-exposed at baseline or seroconverted due to an infection during the trial, sera were also tested at each time point for antibodies against the SARS-CoV-2 N protein using an ELISA (EI 2606-9601-2 G, EUROIMMUN Medizinische Labordiagnostika AG, Lübeck, Germany), since the vaccine does not contain the N protein. Participants who seroconverted during trial (after baseline) were excluded from statistical analysis.

## 3. Results

### 3.1. Response to the Third Dose in SARS-CoV-2-Naïve Participants

In the SARS-CoV-2-naïve participants, the neutralizing antibody GMTs against wild-type virus and the Delta variant increased on D85 and D208 from the pre-dose levels on D43, following a third dose on D57 and D180, respectively (Figure 1, Table 1). The seroconversion rates and GMFRs were higher after the third doses than after the first two doses (Table 1). 

In participants aged >60 years, GMTs against both wild-type and Delta on D208 after the third dose on D180 were higher than those on D85 after the third dose on D57, although the 95% CIs overlapped (Figure 1, Table 1). Seroconversion rates were higher after the third dose on D180 than after the third dose on D57, which was also reflected in the higher GMFRs from baseline titers suggesting a better response to the later dose (Table 1).

The GMTs and GMFRs were higher in participants aged 18–60 years than in those >60 years of age after the third dose on D180, although the 95% CIs were overlapping. The seroconversion rates for Delta were 100% for all participants, irrespective of age, and 100% and 90% for wild-type in participants aged 18–60 years and >60 years, respectively (Table 1). 

### 3.2. Response to the Third Dose in Participants Pre-Exposed to SARS-CoV-2

Prior to the third dose on D180, the GMTs against wild-type and Delta were still detectable in the nine pre-exposed participants, but not in the naïve participants. On D208, the point estimates for GMTs against wild-type and Delta were higher in the pre-exposed participants than those in naïve participants (Figure 1B). The data for the two pre-exposed participants who received a third dose on D57 are included for completeness, and show a tendency to higher GMTs compared with the naïve participants at the time points shown (Figure 1A).

## 4. Discussion

We recently reported that three doses of CVnCoV (12 µg) had an acceptable safety profile in both young and older adults, and induced an increase in IgG and neutralizing antibody titers against the SARS-CoV-2 wild-type [11]. Here, we compared the neutralizing antibody responses to the wild-type and Delta variant induced after a third dose of CVnCoV. 

Four weeks after the third dose administered on D57 or D180, neutralizing antibody GMTs increased against both the wild-type and Delta variant in SARS-CoV-2-naïve participants above the levels observed on D43 after the first two doses. This demonstrates that the first two doses of CVnCoV induced immune memory. The neutralizing antibody GMTs against Delta were lower than those against wild-type on D43 after the two doses, but reached similar levels, or higher as those for the wild-type after the third dose, demonstrating that a robust immune response against Delta variant was induced. These findings are consistent with other studies showing that homologous and heterologous mRNA booster vaccines increased immune responses against SARS-CoV-2 variants [13]. 

In participants aged >60 years, the D180 dose induced higher GMTs (although with overlapping 95% CIs) and seroconversion rates for Delta than the D57 dose. The results suggest that a third dose administered at a later time-point is potentially more immunogenic than the earlier third dose, at least in individuals aged >60 years.

We compared the immune responses to CVnCoV by age, because older individuals are at a higher risk of serious SARS-CoV-2 disease. In younger participants, GMTs for wild-type and Delta were about 2-fold higher than those in adults aged >60 years after 2 doses of CVnCoV and about 2.5-fold higher after the D180 dose. However, the GMTs were higher after three doses than after two in both younger and older individuals, suggesting all of the participants benefitted from the third dose on D180. Immune responses are known to decline with age due to immunosenescence, and this could explain the age-related observations in our study, which have also been shown for other COVID-19 vaccines [4,14].

We previously reported that two doses of CVnCoV induced seroconversion in more than 60% of individuals who were immunologically naïve for SARS-CoV-2, although GMTs waned to baseline levels on D180 in our phase 1 and 2a studies [7,11]. Here, we observed that SARS-CoV-2-pre-exposed individuals still had measurable neutralizing antibodies against both wild-type and Delta up to six months after the first two doses. This indicates that two doses of CVnCoV induced a more potent and longer-lasting immune response against the wild-type and Delta variant in pre-exposed individuals, compared with SARS-CoV-2-naïve individuals. The participants who were naturally pre-exposed by a SARS-CoV-2 infection and who received the D180 booster had the highest D208 wild-type and Delta neutralizing response compared to those who were naïve at baseline (non-overlapping CIs). 

In conclusion, a third CVnCoV dose induced strong neutralizing antibody responses against the SARS-CoV-2 wild-type virus in adults aged 18–60 and >60 years, demonstrating that the first two doses induced immune memory. The third dose induced similar levels of neutralizing responses against the wild-type virus and the Delta variant in both naïve and pre-exposed participants. This is in alignment with the current knowledge from licensed COVID-19 vaccines that a third dose is beneficial against SARS-CoV-2 variants [15,16,17].

Although the development of CVnCoV has stopped, these cross-neutralizing immune responses against SARS-CoV-2 variants are promising for the next-generation SARS-CoV-2 vaccines, which are based on the same mRNA platform and are being optimized for variants [18].

## Figures and Tables

**Figure 1 vaccines-10-00508-f001:**
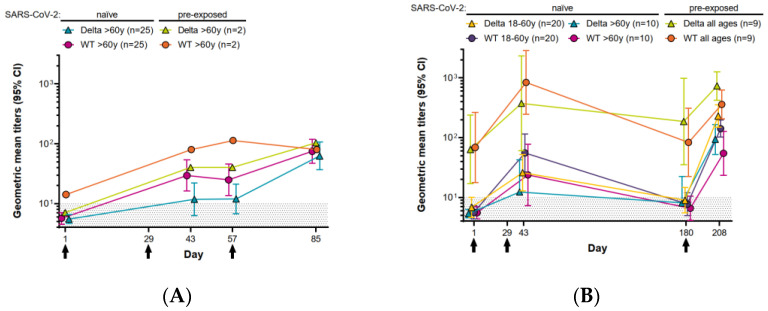
Neutralizing antibody response against wild-type SARS-CoV-2 virus and Delta variant after CVnCoV vaccination. The geometric mean titers (GMTs) of neutralizing antibodies with 95% confidence intervals (CIs) to wild-type (WT; filled circles) and Delta (filled triangles) are shown. (**A**) Participants aged >60 years with a third dose on Day (D)57 and (**B**) participants aged 18–60 years and >60 years with a third dose on D180. SARS-CoV-2-naïve (seronegative for N protein throughout the trial period) and SARS-CoV-2-pre-exposed (seropositive for N protein at baseline) participants are indicated by color. The black arrows indicate the days of vaccinations: D1 (baseline), D29 (**A**,**B**) and D57 (**A**) or D180 (**B**)). *n* = number of participants with available data per group for all time points. Values above the black-dotted region (≥10) were considered positive.

**Table 1 vaccines-10-00508-t001:** Geometric mean titers, seroconversion rates and geometric mean fold rises of Delta and wild-type neutralizing antibodies 28 days after 2 (D43) and 3 (D85 and D208) CVnCoV doses in SARS-CoV-2-naïve participants.

	D57 Dose >60 Years		D180 Dose >60 Years		D180 Dose 18–60 Years	
	Wild-Type	Delta	Wild-Type	Delta	Wild-Type	Delta
GMT (95% CI)						
D43	29.5 (16.3–53.5)	11.8 (6.3–22.1)	23.8 (7.3–77.5)	12.3 (3.6–42.6)	55.6 (26.8–115.2)	25.9 (12.8–52.5)
D85	74.6 (46.9–118.9)	62.8 (36.8–107.2)	-	-	-	-
D208	-	-	54.6 (23.3–128.4)	93.5 (52.4–167.0)	141.7 (101.9–197.1)	230.2 (148.8–356.1)
Seroconversion ^a^, *n*/N (%)						
D43	16/25 (64%)	5/25 (20%)	5/10 (50%)	2/10 (20%)	15/20 (75%)	10/20 (50%)
D85	23/25 (92%)	20/25 (80%)	-	-	-	-
D208	-	-	9/10 (90%)	10/10 (100%)	20/20 (100%)	20/20 (100%)
GMFR ^b^						
D43	5.2	2.1	4.3	2.3	9.8	3.8
D85	13.2	11.4	-	-	-	-
D208	-	-	9.8	17.4	25.1	33.7

^a^ Defined as at least a four-fold increase in titers over baseline; ^b^ geometric mean fold rises of titers over baseline values; CI = confidence interval; D = day; GMFR = geometric mean fold rise; GMT = geometric mean titers; and *n*/N = number of participants/total number of participants in the subset.

## Data Availability

The data presented in this study are available on request from the corresponding author.

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
