# Peer review of "A Third Dose of the COVID-19 Vaccine, CVnCoV, Increased the Neutralizing Activity against the SARS-CoV-2 Wild-Type and Delta Variant"

_vaccines, 2022, doi:10.3390/vaccines10040508_

Round 1

Reviewer 1 Report

The article “Third dose of COVID-19 vaccine, CVnCoV, increased neutralizing activity against SARS-CoV-2 wild-type and Delta variant” describes the neutralization antibody response to the wild type and Delta variants induced after a third dose of a former candidate mRNA vaccine against SARS-CoV-2 on day 57 and 180.

The aim of the study and the results are quite interesting and could be useful for future next generation SARS-CoV-2 vaccines.

The article in some points is not easy to be immediately understood. I think that the study design should be better described and clarified in a broader way.

The study design is referred to a previous paper not published, but just submitted. I think that would be more correct make a summary in this paragraph, since the previous paper has not been still published.

Why did you choose one group with people over 60 and another group with people aged 18-60? I think it would have much more sense to make the experiments on the same groups in order to have perfectly comparable results.

As regards the pre-exposed patients is not clear if they exposed during the experiment period or before. If before, why did you choose just 2 and 9 people in the two experiments? However, 2 people is not a significant number to make some hypotheses or conclusions. You should increase the number.

Line 96-97: I think there is a mistake. You are speaking of wild type and not of delta variant for the data you indicated.

The Figure 1 is not really easy to interpreter. I think it’s better to use different colours for the different categories and not different symbols.

Delete lines 130-133: If you cannot show data it’s better to delete this part.

Author Response

The article “Third dose of COVID-19 vaccine, CVnCoV, increased neutralizing activity against SARS-CoV-2 wild-type and Delta variant” describes the neutralization antibody response to the wild type and Delta variants induced after a third dose of a former candidate mRNA vaccine against SARS-CoV-2 on day 57 and 180.

The aim of the study and the results are quite interesting and could be useful for future next generation SARS-CoV-2 vaccines.

Thank you. Please note that the changes we made in the manuscript are in track changes mode. The line numbers mentioned here are related to the ‘simple markup’ option in MS-Word and do not match the firs submitted manuscript.

The article in some points is not easy to be immediately understood. I think that the study design should be better described and clarified in a broader way.

To better describe the study design and describe the scope of the article we added text to the introduction in lines 52-56

The study design is referred to a previous paper not published, but just submitted. I think that would be more correct make a summary in this paragraph, since the previous paper has not been still published.

Text was added to better describe the study design and the subset of participants analysed in section 2.1. ‘Study Design and Participants’ lines 61-63 and 66-68 of this manuscript.

Why did you choose one group with people over 60 and another group with people aged 18-60? I think it would have much more sense to make the experiments on the same groups in order to have perfectly comparable results.

The third dose at day 57 was investigated in subjects > 60 years of age only because at the time point of study design and performance, it was important to consider whether older people might need a 3rd dose in order to rapidly mount an immune-response of sufficient magnitude in the face of the pandemic. .

As regards the pre-exposed patients is not clear if they exposed during the experiment period or before. If before, why did you choose just 2 and 9 people in the two experiments? However, 2 people is not a significant number to make some hypotheses or conclusions. You should increase the number.

In the legend of Figure 1 it is described that SARS-CoV-2 pre-exposure is defined as seropositive for N protein at baseline, so that it is clear for the reader that these participants seroconverted already before administration of the first CVnCoV dose. Furthermore we added text to the methods part of the manuscript, see lines 85-86 and 89-90 for better clarification.

SARS-CoV-2 pre-exposure was determined retrospectively, therefore recruitment of n=2 (for D57 booster) and n=9 (for D180 booster) pre-exposed subjects was stochastically and not defined by the trials’ design.

We agree that the 2 participants with D57 booster who were pre-exposed is too low a number to make significant conclusions as we described in lines 113-115 that these have only been included for completeness. Nevertheless it is of note that these 2 pre-exposed subjects showed highest neutralizing response against Delta already after 2 doses (see D43 & D57 time points) compared to the naïve. As explained, it was not designed to recruit seropositive subjects and this was assessed only retrospectively.

Line 96-97: I think there is a mistake. You are speaking of wild type and not of delta variant for the data you indicated.

Thank you. The text was changed accordingly, see now lines 104-107.

The Figure 1 is not really easy to interpreter. I think it’s better to use different colours for the different categories and not different symbols.

We changed the Figure 1 symbols and colour scheme and modified the legend accordingly. Furthermore values at one time point are now displayed with a slight horizontal offset to ease comparisons of the CIs. 

Delete lines 130-133: If you cannot show data it’s better to delete this part.

Text (current version line137) “After the D57 dose, neutralizing immune responses against variants of concern, other than Delta, were also increased above those observed after the two doses (data not shown).” was removed from the manuscript.

Reviewer 2 Report

In this work, authors show the effect of a third dose of COVID-19 vaccine, CVnCoV, on the increase of neutralizing activity against SARS-CoV-2 wild-type and Delta variant, for two different age-based groups, at two different times (day 57 and 180).

Several points have to be discussed:

-Geometric mean titers (GMT) of neutralizing antibodies in the 18-60 interval immunized at D57 naïve. Is there any data about this group?

-After the D57 dose, neutralizing immune responses against variants of concern, other than Delta, were also increased above those observed after the two doses (data not shown). Show this data or remove this sentence.

Several comparisons are made/discussed in this work:

GMTs at Day 57 vs Day 180

GMTs based on age group

In discussion, I miss the comparison between naïve and pre exposed GMT values after the third dose. Is it really an enhancement of GMT in pre-exposed individuals?

Author Response

In this work, authors show the effect of a third dose of COVID-19 vaccine, CVnCoV, on the increase of neutralizing activity against SARS-CoV-2 wild-type and Delta variant, for two different age-based groups, at two different times (day 57 and 180).

Several points have to be discussed:

Thank you. Please note that the changes we made in the manuscript are in track changes mode. The line numbers mentioned here are related to the ‘simple markup’ option in MS-Word and do not match the firs submitted manuscript.

-Geometric mean titers (GMT) of neutralizing antibodies in the 18-60 interval immunized at D57 naïve. Is there any data about this group?

We replied to a similar questions from the other reviewer:

Data was not collected from 18-60 years of age naïve and immunized with a third dose on D57. The third dose at day 57 was investigated in subjects > 60 years of age only because at the time point of study design and performance, it was important to consider whether older people might need a 3rd dose to rapidly mount an immune response of sufficient magnitude in the face of the pandemic.

-After the D57 dose, neutralizing immune responses against variants of concern, other than Delta, were also increased above those observed after the two doses (data not shown). Show this data or remove this sentence.

Delete lines 130-133: If you cannot show data it’s better to delete this part.

Text (current version line137) “After the D57 dose, neutralizing immune responses against variants of concern, other than Delta, were also increased above those observed after the two doses (data not shown).” was removed from the manuscript.

Several comparisons are made/discussed in this work:

GMTs at Day 57 vs Day 180

GMTs based on age group

In discussion, I miss the comparison between naïve and pre exposed GMT values after the third dose. Is it really an enhancement of GMT in pre-exposed individuals?

We added text to further discuss the comparison between naïve and pre exposed GMT values after the third dose in lines 168-171.

Round 2

Reviewer 1 Report

The authors made all the changes I asked and clarified some parts of the manuscript.

I think that now the article is suitable for publication.